# Performance of Screening Strategies for Latent Tuberculosis Infection in Patients with Inflammatory Bowel Disease: Results from the ENEIDA Registry of GETECCU

**DOI:** 10.3390/jcm11133915

**Published:** 2022-07-05

**Authors:** Sabino Riestra, Carlos Taxonera, Yamile Zabana, Daniel Carpio, María Chaparro, Jesús Barrio, Montserrat Rivero, Antonio López-Sanroman, María Esteve, Ruth de Francisco, Guillermo Bastida, Santiago García-López, Miriam Mañosa, María Dolores Martin-Arranz, José Lázaro Pérez-Calle, Jordi Guardiola, Fernando Muñoz, Laura Arranz, José Luis Cabriada, Mariana Fe García-Sepulcre, Mercè Navarro, Miguel Ángel Montoro-Huguet, Elena Ricart, Fernando Bermejo, Xavier Calvet, Marta Piqueras, Esther Garcia-Planella, Lucía Márquez, Miguel Mínguez, Manuel Van Domselar, Luis Bujanda, Xavier Aldeguer, Beatriz Sicilia, Eva Iglesias, Guillermo Alcaín, Isabel Pérez-Martínez, Valeria Rolle, Andrés Castaño-García, Javier P. Gisbert, Eugeni Domènech

**Affiliations:** 1Gastroenterology Department, Hospital Universitario Central de Asturias, Instituto de Investigación Sanitaria del Principado de Asturias (ISPA), 33011 Oviedo, Spain; ruthdefrancisco@gmail.com (R.d.F.); ipermar_79@hotmail.com (I.P.-M.); bioestadistica@ispasturias.es (V.R.); castaogarcia@gmail.com (A.C.-G.); 2Gastroenterology Department, Hospital Clínico Universitario San Carlos, Instituto de Investigación del Hospital Clínico San Carlos (IdISSC), 28040 Madrid, Spain; carlos.taxonera@salud.madrid.org; 3Gastroenterology Department, Hospital Universitari Mútua Terrassa, Centro de Investigación Biomédica en Red de Enfermedades Hepáticas y Digestivas (CIBERehd), 08221 Terrassa, Spain; yzabana@gmail.com (Y.Z.); mariaesteve@mutuaterrassa.cat (M.E.); 4Gastroenterology Department, Complexo Hospitalario Universitario de Pontevedra, 36071 Pontevedra, Spain; dcarlop1@yahoo.es; 5Gastroenterology Department, Hospital Universitario de La Princesa, Instituto de Investigación Sanitaria Princesa (IIS-IP), Universidad Autónoma de Madrid (UAM), Centro de Investigación Biomédica en Red de Enfermedades Hepáticas y Digestivas (CIBERehd), 28006 Madrid, Spain; mariachs2005@gmail.com (M.C.); javier.p.gisbert@gmail.com (J.P.G.); 6Gastroenterology Department, Hospital Universitario Río Hortega, 47012 Valladolid, Spain; jbarrioa95@gmail.com; 7Gastroenterology Department, Hospital Universitario Marqués de Valdecilla, Universidad de Cantabria, Instituto de Investigación Sanitaria Valdecilla (IDIVAL), 37008 Santander, Spain; montserrat.rivero@scsalud.es; 8Gastroenterology Department, Hospital Universitario Ramón y Cajal, 28034 Madrid, Spain; mibuzon@gmail.com; 9Gastroenterology Department, Hospital Universitari i Politecnic La Fe, Centro de Investigación Biomédica en Red de Enfermedades Hepáticas y Digestivas (CIBERehd), 46026 Valencia, Spain; guille.bastida@gmail.com; 10Gastroenterology Department, Hospital Universitario Miguel Servet, Instituto de Investigación Sanitaria Aragón (IISA), 50009 Zaragoza, Spain; sgarcia.lopez@gmail.com; 11Gastroenterology Department, Hospital Universitari Germans Trials i Pujol, Centro de Investigación Biomédica en Red de Enfermedades Hepáticas y Digestivas (CIBERehd), 08916 Badalona, Spain; mmanosa.germanstrias@gencat.cat (M.M.); eugenidomenech@gmail.com (E.D.); 12Gastroenterology Department, Hospital Universitario La Paz, Instituto de Investigación Sanitaria del Hospital La Paz (IdiPAZ), 28046 Madrid, Spain; mmartinarranz@salud.madrid.org; 13Gastroenterology Department, Hospital Universitario Fundación Alcorcón, 28922 Madrid, Spain; jlperezc@fhalcorcon.es; 14Gastroenterology Department, Hospital Universitario de Bellvitge, Institut d’Investigació Biomèdica de Bellvitge (IDIBELL), 08907 Barcelona, Spain; jguardiola@bellvitgehospital.cat; 15Gastroenterology Department, Hospital Universitario de Salamanca, 37007 Salamanca, Spain; jmunozn@gmail.com; 16Gastroenterology Department, Hospital Nuestra Señora de la Candelaria, 38010 Tenerife, Spain; lauraarranz@gmail.com; 17Gastroenterology Department, Hospital de Galdakao-Usansolo, 48960 Galdakao, Spain; jcabriada@gmail.com; 18Gastroenterology Department, Hospital General Universitario de Elche, 03203 Alicante, Spain; marifegarciasepulcre@gmail.com; 19Gastroenterology Department, Hospital Sant Joan Despí-Moisès Broggi, 08970 Barcelona, Spain; mnavarrollavat@gmail.com; 20Gastroenterology Department, Hospital San Jorge, 22004 Huesca, Spain; maimontoro@gmail.com; 21Gastroenterology Department, Hospital Clinic i Provincial, IDIBAPS, Centro de Investigación Biomédica en Red de Enfermedades Hepáticas y Digestivas (CIBERehd), 08036 Barcelona, Spain; ericart@clinic.cat; 22Gastroenterology Department, Hospital Universitario de Fuenlabrada, Instituto de Investigación Sanitaria Hospital La Paz (IdiPaz), 28942 Madrid, Spain; fbermejos1@gmail.com; 23Gastroenterology Department, Hospital Universitari Parc Taulí, Depàrtament de Medicina Universitat Autònoma de Barcelona, Sabadell, CIBERehd, 08208 Sabadell, Spain; xavier.calvet.c@gmail.com; 24Gastroenterology Department, Consorcio Sanitario de Terrasa, 08227 Barcelona, Spain; piqueras72@gmail.com; 25Gastroenterology Department, Hospital de la Santa Creu i Sant Pau, 08041 Barcelona, Spain; egarciapl@santpau.cat; 26Gastroenterology Department, Hospital del Mar, Institut Hospital del Mar d’Investigacions Mèdiques, 08003 Barcelona, Spain; lmarquez@parcdesalutmar.cat; 27Gastroenterology Department, Hospital Clínico Universitario de Valencia, 46010 Valencia, Spain; miguel.minguez@uv.es; 28Gastroenterology Department, Hospital Universitario de Torrejón, 28850 Madrid, Spain; manuelvandomselaar@yahoo.com; 29Gastroenterology Department, Hospital Universitario de Donostia/Biodonostia, Universidad del País Vasco (UPV/EHU), Centro de Investigación Biomédica en Red de Enfermedades Hepáticas y Digestivas (CIBERehd), 20014 San Sebastián, Spain; luis.bujandafernandezdepierola@osakidetza.net; 30Gastroenterology Department, Hospital Universitario de Girona Dr Josep Trueta, 17007 Girona, Spain; doctoraldeguer@yahoo.es; 31Gastroenterology Department, Hospital Universitario de Burgos, 09006 Burgos, Spain; bsicilia4@gmail.com; 32Gastroenterology Department, Hospital Reina Sofía, IMIBIC, 14004 Córdoba, Spain; evaiflores@gmail.com; 33Gastroenterology Department, Hospital Clínico de Málaga, 29010 Málaga, Spain; galcainm@hotmail.com

**Keywords:** inflammatory bowel disease, latent tuberculosis infection, tuberculin skin test, interferon gamma release assays

## Abstract

(1) Aims: Patients receiving antitumor necrosis factor (anti-TNF) therapy are at risk of developing tuberculosis (TB), usually due to the reactivation of a latent TB infection (LTBI). LTBI screening and treatment decreases the risk of TB. This study evaluated the diagnostic performance of different LTBI screening strategies in patients with inflammatory bowel disease (IBD). (2) Methods: Patients in the Spanish ENEIDA registry with IBD screened for LTBI between January 2003 and January 2018 were included. The diagnostic yield of different strategies (dual screening with tuberculin skin test [TST] and interferon-ץ-release assay [IGRA], two-step TST, and early screening performed at least 12 months before starting biological treatment) was analyzed. (3) Results: Out of 7594 screened patients, 1445 (19%; 95% CI 18–20%) had LTBI. Immunomodulator (IMM) treatment at screening decreased the probability of detecting LTBI (20% vs. 17%, *p* = 0.001). Regarding screening strategies, LTBI was more frequently diagnosed by dual screening than by a single screening strategy (IGRA, OR 0.60; 95% CI 0.50–0.73, *p* < 0.001; TST, OR 0.76; 95% CI 0.66–0.88, *p* < 0.001). Two-step TST increased the diagnostic yield of a single TST by 24%. More cases of LTBI were diagnosed by early screening than by routine screening before starting anti-TNF agents (21% [95% CI 20–22%] vs. 14% [95% CI 13–16%], *p* < 0.001). The highest diagnostic performance for LTBI (29%) was obtained by combining early and TST/IGRA dual screening strategies in patients without IMM. (4): Conclusions: Both early screening and TST/IGRA dual screening strategies significantly increased diagnostic performance for LTBI in patients with IBD, with optimal performance achieved when they are used together in the absence of IMM.

## 1. Introduction

Inflammatory bowel diseases (IBD) make up a group of chronic inflammatory conditions including Crohn’s Disease (CD), Ulcerative Colitis (UC) and unclassified colitis (IBD unclassified). Treatment with antitumor necrosis factor biologics (anti-TNFs) is important for controlling inflammatory activity in these patients. Patients receiving anti-TNF therapy are at risk of developing tuberculosis (TB), usually due to the reactivation of a latent TB infection (LTBI). Following the first report of this treatment-related complication [1], several scientific societies and international organizations published guidelines for the screening and treatment of LTBI in patients receiving biological treatment [2]. The adherence to these recommendations led to a decrease, but not the complete disappearance, in the number of cases of active TB [3,4,5]. Active TB still occurs, probably due to inadequate compliance with the screening recommendations [6], the limitations of immunodiagnostic tests [7] and the occurrence of *de novo* infections [4]. Recently, the risk of active TB during anti-TNF treatment has been related to the local epidemiology of TB and, additionally, it has been reported that up to 70% of cases occur in patients with negative baseline LTBI screening [8].

The low sensitivity of LTBI screening tests (tuberculin skin test (TST) and the interferon-γ-release assay (IGRA)) when performed under immunosuppressive therapies [7,9] has been posed as one of the potential factors that increase the risk of active TB during anti-TNF treatment. Different strategies have been developed to improve the performance of immunodiagnostic tests. Early screening, performed ideally at inflammatory bowel disease (IBD) diagnosis or in the absence of immunosuppressive drugs [9,10], increases its sensitivity for detecting LTBI. Adding a second TST seven to 14 days after a first negative test increases the LTBI diagnostic yield by 5–25% [10,11,12]. Finally, dual screening (TST plus IGRA) also increases the sensitivity for detecting LTBI compared to a single screening test (TST or IGRA), in patients with rheumatic diseases [13,14] or IBD [12]. However, there is a lack of studies assessing the best strategies in different clinical settings.

The European Crohn’s and Colitis Organisation (ECCO) recommends adapting LTBI diagnostic and treatment strategies to the local TB epidemiology in anti-TNF candidates [15]. In 2018 Spain was considered, for the first time, a country with a low incidence of TB (9.4 cases/100,000 inhabitants/year), according to WHO criteria [16], although with wide variations in incidence among different Spanish regions [17]. In 2014, it was estimated that the prevalence of LTBI in Spain was 15% (95% CI 6.3–27%) [18], whereas the data reported for IBD patients was highly variable (13–34%) [10,11,12]. Taking these data into account, the Spanish Working Group on Crohn’s Disease and Ulcerative Colitis (GETECCU) published its updated recommendations for the screening and treatment of LTBI in 2021 [19] and recommended single screening (TST) for immunocompetent patients and dual screening (TST/IGRA) for immunocompromised patients.

The main objective of this study was to describe the prevalence and risk factors of LTBI and to evaluate the diagnostic yield of different LTBI screening strategies (TST/IGRA dual screening, single screening, two-step TST, routine screening and early screening) in a large database of IBD patients.

## 2. Methods

This is a retrospective, descriptive, multi-center study, conducted in patients included in the ENEIDA registry. ENEIDA is a nationwide Spanish database of IBD patients promoted by GETECCU, which is a non-profit association whose mission is to improve the lives of people with IBD by promoting excellence in health care, teaching and research and influencing political and social initiatives. [20]. The registry was approved by the Ethics Committee of each participating center and all patients gave their informed consent. The study was approved by the ENEIDA Scientific Committee and, at the time of the data extraction (January 2018), the registry included 31,827 patients from 33 centres (Appendix A).

### 2.1. Study Population and Definitions

We included all patients who had been screened for LTBI from January 2003 (the date of the publication of the first GETECCU recommendations for LTBI screening in IBD patients) [21] to January 2018. Those patients who had previously been diagnosed with active TB or LTBI were excluded. We considered that a patient had undergone LTBI screening when a TST and/or IGRA had been performed and/or a recent contact with a bacilliferous subject was identified. In patients with more than one LTBI screening test recorded in the database, the result of the first screening was taken into account to evaluate the frequency of LTBI. A patient was considered to have LTBI if there was a positive result in either screening test (positive IGRA and/or TST > 5 mm), when there were lesions suggestive of an old TB infection on a chest X-ray (calcified nodes or nodules, pleural apical thickening, fibrous tracts), or when recent contact with a bacilliferous TB patient was identified [19].

### 2.2. Performance of Tests and Screening Strategies for Latent Tuberculosis Infection

We analyzed the overall frequency of LTBI, as well as the diagnostic yield per single test (IGRA or TST). We also evaluated the association of a positive screening with demographic and clinical variables including gender, age at diagnosis of IBD, age at LTBI screening, type of IBD, smoking habit, use of immunomodulators (IMM) (thiopurines, methotrexate or calcineurinics) and anti-TNF (Infliximab, adalimumab or golimumab) at screening, autonomous community of residence (Madrid, Aragón, Asturias, Canarias, Cantabria, Castilla-León, Catalonia, Basque country, Valencia), country of origin (classified according to WHO criteria as high, intermediate or low incidence of TB) [14] and the year in which the screening was performed.

Subsequently, we evaluated the diagnostic yield of different screening strategies. First, we compared TST/IGRA dual screening to single screening (TST or IGRA); secondly, we evaluated the strategy of a second TST (two-step TST) in patients with a negative first TST. Finally, we compared early screening (considered as when screening tests were performed at least 12 months before starting biological treatment) to routine screening (when tests were done in the three months before initiating biological treatment).

### 2.3. Statistical Analysis

Categorical variables are reported as absolute and relative frequencies (number, percentage, and 95% confidence interval). Continuous variables are expressed as the mean and standard deviation or median and interquartile range (IQR), as appropriate. The initial comparability between groups was analyzed with the chi square test or Fisher’s exact test for qualitative variables and with the Student’s *t* test or median test for quantitative data. To assess the performance of different tests and diagnostic strategies for LTBI, we used a logistic regression model (with the presence of LTBI as a dependent variable and the independent variables mentioned in the previous section). To assess the performance of TST/IGRA dual screening compared to a single screening, we used a logistic regression model (the presence of LTBI as a dependent variable, while leaving only the screening strategy as a separate variable). To evaluate the performance of early compared to routine screening, the diagnostic yield was calculated for each strategy and compared using a difference estimation test for proportions. Crude and adjusted odds ratios with their respective 95% confidence intervals (CIs) were reported. A *p*-value of <0.05 was considered to be statistically significant. In addition, the non-inferiority criterion was evaluated for subgroup analysis comparisons showing a non-significant trend. The non-inferiority criterion was not met for any comparisons if the lower limit of the 95% CI was lower than the pre-specified non-inferiority criterion of ≤0.65. All analysis were performed with R software version 4.0.2.

## 3. Results

In total, 7594 IBD patients screened for LTBI between January 2003 and January 2018 were included. The demographic and clinical characteristics of patients are summarized in Table 1.

Overall, 1445 patients (19%; 95% CI 18–20%) were diagnosed with LTBI. The frequency of LTBI, regarding the tests performed for screening and the use of IMM at the time the screening was performed, are shown in Table 2. In 369 patients, the first LTBI screening collected in ENEIDA was performed during anti-TNF treatment; LTBI was less frequently detected among patients taking anti-TNF than in patients without such treatment (12% [95% CI 8.6–16%] vs. 19% [95% CI 19–20%], respectively; *p* < 0.001).

Table 3 shows the factors associated with detecting LTBI. Age at screening, male gender, smoking habit, autonomous community of residence, year in which the screening was performed, origin in countries with a high incidence of active TB (≥40 cases/10^5^ inhabitants/year), and lack of IMM and anti-TNF treatment at screening, were all associated with higher rates of detected LTBI.

When we exclude patients undergoing anti-TNF at screening (*n* = 369) from the logistic regression analysis, the factors associated with LTBI diagnosis were the same as for the general cohort (Appendix A).

Overall, LTBI was more common in men than in women (21% vs. 17%, *p* = 0·01), although these differences were only observed in people over 55 years of age (Appendix A). Since 2003, a progressive decrease in LTBI frequency (slope: −1.6; 95% CI −2.2 to −1.0; *p* = 0.013) was observed (Appendix A). Moreover, a decreasing use of TST and an increasing use of IGRA were observed over time. While TST and IGRA were performed in 91% and 22% of the screenings performed from 2009 to 2011, these figures were 63% and 62%, respectively, in the 2015–2017 period.

### 3.1. Latent Tuberculosis Infection Screening Strategies

#### 3.1.1. TST/IGRA Dual Screening

In total, 1471 and 6068 patients were screened using a dual or a single screening strategy, respectively. Overall, the likelihood of a positive screening was significantly higher with the dual screening strategy than with a single screening. The probability of diagnosing LTBI with a single IGRA decreased by 40% (OR: 0.60; 95% CI 0.50–0.73, *p* < 0.001), and, in the case of a single TST, this probability decreased by 24% (OR: 0.76; 95% CI 0.66–0.88, *p* < 0.001), compared to TST/IGRA dual screening (Table 4). Significant differences were maintained for the subgroup of patients without IMM treatment. In the subgroup of patients with IMM treatment, a non-significant trend was observed towards a lower sensitivity of a single IGRA (*p* = 0.2) and of a single TST (*p* = 0.2) when compared with dual screening, with none of the single tests achieving the non-inferiority criterion compared to TST/IGRA dual testing (Table 4). In patients screened with both TST and IGRA (*n* = 1471), concordance between tests was low (kappa 0.49; 95% CI 0.42–0.55).

#### 3.1.2. Screening with Two-Step TST (Booster)

Eight hundred and two out of 6028 patients (13%) had a first positive TST (one-step strategy). A second TST was performed on 2332 of the 5226 patients (45%) with a first negative TST and was positive in 163 (6%). The two-step TST strategy increased the likelihood of detecting LTBI by 24% (OR 1.24; 95% CI 1.12–1.37, *p* < 0.001) (Table 4).

#### 3.1.3. Early Screening

In total, 4365 and 1421 patients were screened using an early or routine screening strategy, respectively. Overall, LTBI was diagnosed more frequently using early than routine screening (21% vs. 14%, *p* < 0.001). Routine screening was associated with a 37% reduction in the likelihood of detecting LTBI vs. early screening (OR 0.63; 95% CI 0.53–0.74; <0.001) (Table 4). Significant differences remained for the subgroup of patients without IMM. In the subgroup of patients on IMM treatment, a non-significant trend was observed towards a lower LTBI detection capacity of routine screening compared to early screening (OR 0.78; 95% CI 0.60–1.03; *p* = 0.08) (Table 4).

#### 3.1.4. Interactions between Early Screening and Dual Screening Strategies

The highest LTBI detection capacity (188 of 643 patients [29%]) was obtained with the association of the early screening and TST/IGRA dual screening strategies in patients without IMM (Table 4).

#### 3.1.5. Probability of LTBI according to Demographic, IMM Use, and Screening Strategy

Table 5 shows the probability of detecting LTBI according to gender, smoking status, IMM use at screening, screening strategy used and origin in a country with a high or intermediate/low incidence of TB. Thus, the probability of LTBI ranged from 53% (men, ever smoker and origin in a country with a high incidence of TB, when a dual and early screening was used while without IMM) to 8% (women, never smoker and origin in a country with an intermediate/low incidence, when a single IGRA screening was used while on IMM). The results obtained when patients receiving anti-TNF treatment at the time of LTBI screening were excluded are similar to those of the overall series (Appendix A).

## 4. Discussion

To date, the best strategy for the LTBI screening of patients for whom biological treatment is planned remains unknown, and should always depend on the local epidemiology of TB (incidence of active TB and prevalence of LTBI) [15]. In this study, the frequency of LTBI in a large cohort of Spanish patients with IBD was 19%, with a steady decline over a 15-year period (25% in 2003 to 18% in 2017). During this period, the incidence of active TB among the background Spanish population also dropped from 27 to 9.4 cases/10^5^ inhabitants/year, leading Spain to become a country with a low incidence of TB (<10 cases/10^5^ inhabitants/year) [22]. In our study, age, male gender, smoking status and being from countries with a high incidence of TB were independently associated with a higher frequency of LTBI, as was previously reported [23]. Moreover, we observed marked differences in LTBI prevalence between autonomous communities, some of which are geographically close. In this regard, an autonomous community with a higher proportion of people over the age of 65 (i.e., Castilla and León) or a larger immigrant population (i.e., Catalonia) may explain the higher frequency of LTBI in some regions [24].

ECCO recommends performing LTBI screening at the time of IBD diagnosis, before starting immunosuppressive therapy and, preferably, when a low inflammatory load is present [15]. At present, only a few patients will have the screening performed under these optimal conditions and no studies have analyzed the benefits of this strategy. The study evaluated in a real-life setting the sensitivity of early screening performed on patients for whom anti-TNF therapy was not foreseen compared with routine screening performed before the initiation of biological therapy, and we found that early screening markedly increased the likelihood of detecting LTBI in both UC and CD patients. It has been reported that the sensitivity of TST was lower when it was indicated as part of the mandatory screening for LTBI before the initiation of biological therapy, and this has been related to an increased use of steroids and IMM and to a higher systemic inflammatory load associated with active IBD [10]. Our results show for the first time that routine screening affects not only the sensitivity of TST, but also that of IGRA and TST/IGRA dual testing.

Early screening increased LTBI diagnostic capacity by 43% in the cohort of patients who were not receiving IMM compared to routine screening, whereas this increase was smaller among patients on IMM. In the same way, the incremental benefit of TST/IGRA dual screening over single screening was more marked in those patients who were not receiving IMM than in those on IMM. The IMM subgroup analysis for the diagnostic performance of early screening and dual screening reinforces the evidence from the multivariate analysis that IMM had a pronounced negative effect on the diagnostic yield of screening tests for LTBI in all situations.

In our study, the best diagnostic yield for LTBI was obtained with the simultaneous use of early and TST/IGRA dual screening strategies. The incremental benefit of dual testing was higher when combined with an early screening strategy compared to when routine screening was performed. This could be explained by the fact that, at that time, a greater proportion of patients with routine screening were receiving IMM drugs (44% vs. 19%). Therefore, the results of our study strongly support the need for LTBI screening when a patient is not receiving IMM therapy.

The limitations of this study include its retrospective design, which prevented the analysis of the possible influence of corticosteroid treatment and the inflammatory load of IBD at the time of LTBI screening. Moreover, the sample sizes of some subgroup analyses may diminish the significance of our findings. In addition, our results should not be extrapolated to other geographical areas with a different local TB epidemiology. Conversely, the main strengths of the study are the inclusion of a large number of patients and the fact that information on LTBI screening strategies used in real life in a nationwide multi-center setting was systematically collected.

In conclusion, in an area with an intermediate incidence of TB, both early screening and TST/IGRA dual screening strategies increased the performance of LTBI in patients with IBD. The sum of both strategies maximizes the probability of diagnosing LTBI. Since new cases of active TB still occur in IBD patients on biologic therapy despite preventive actions, it is crucial to screen patients early in the absence of IMM treatment and take advantage of the incremental benefit of associating diagnostic tests for LTBI.

## Figures and Tables

**Table 1 jcm-11-03915-t001:** Clinical and demographic characteristics of the cohort of patients with IBD who were screened for LTBI.

Number of patients screened for LTBI, *n*	7594
Male gender, *n* (%)	3988 (53)
Age at diagnosis of IBD, years, mean (±SD)	36.3 ± 15.2
Type of IBD, *n* (%) - Crohn´s disease - Ulcerative colitis - IBD unclassified	4622 (61) 2813 (37) 159 (2)
Smoking status, *n* (%) - ever - never - unknown	3677 (48) 3377 (45) 540 (7)
Age at LTBI screening, years, mean (± SD)	45.2 ± 15.2
Time between diagnosis of IBD and screening, years, median (IQR)	2.09 (0.19–10.46)
IMM at screening, *n* (%) Azathioprine Mercaptopurine Methotrexate Other IMM	2143 (28) 1703 185 146 109
Anti-TNF at screening, *n* (%)	369 (4.9)
Country of origin with TB incidence ≥ 40	294 (3.9)

LTBI: latent tuberculosis infection; IBD: inflammatory bowel disease; SD: standard deviation; IQR: interquartile range; IMM: immunomodulator; anti-TNF: antitumor necrosis factor; TB: tuberculosis.

**Table 2 jcm-11-03915-t002:** Frequency of LTBI and results of the screening methods.

	Patients Screened, *n*	Patients with LTBI, *n* (%) [95% CI]	*p* Value *
Any test of LTBI ** - no IMM at screening - IMM at screening	7594 5451 2143	1445 (19) [18,19,20] 1086 (20) [19,20,21] 359 (17) [15,16,17,18]	0.001
TST - no IMM at screening - IMM at screening	6028 4234 1794	965 (16) [15,16,17] 736 (17) [16,17,18,19] 229 (13) [11,12,13,14]	<0.001
IGRA - no IMM at screening - IMM at screening	2982 2208 774	386 (13) [12,13,14] 314 (14) [13,14,15,16] 72 (9.3) [7.4–12]	<0.001
Chest X-ray	2365	55 (2.3) [1.8–3]	
Recent contact with baciliferous subject	7594	213 (2.8) [2.5–3.2]	

LTBI: latent tuberculosis; CI: confidence interval; IMM: immunomodulator; TST: tuberculin skin test; IGRA: interferon gamma release assay; * *p* value comparing results of screening performed during IMM and no IMM therapy; ** TST and/or IGRA positive and/or abnormal chest X-ray and/or recent contact with baciliferous subject.

**Table 3 jcm-11-03915-t003:** Factors associated with diagnosis of LTBI in patients with IBD.

	Univariable Analysis			Multivariable Analysis		
	Odds Ratio	95% CI	*p*	Odds Ratio	95% CI	*p*
Gender						
Female	Reference			Reference		
Male	1.22	1.09–1.37	0.001	1.21	1.07–1.37	0.002
Age at TB screening, years	1.04	1.03–1.04	<0.001	1.04	1.04–1.04	<0.001
Type of IBD						
IBD-unclassified	Reference			Reference		
Ulcerative colitis	0.84	0.57–1.25	0.37	0.78	0.52–1.19	0.23
Crohn’s disease	0.88	0.60–1.31	0.50	0.89	0.60–1.36	0.58
Smoking status						
Never	Reference			Reference		
Ever	1.58	1.40–1.78	<0.001	1.53	1.35–1.74	<0.001
Unknown	1.02	0.79–1.30	0.86	0.87	0.66–1.13	0.29
IMM at TB screening						
No	Reference			Reference		
Yes	0.80	0.70–0.91	0.001	0.81	0.70–0.93	0.004
Anti-TNF at TB screening						
No	Reference			Reference		
Yes	0.55	0.39–0.75	<0.001	0.56	0.40–0.77	<0.001
Geographical area						
Madrid	Reference			Reference		
Aragón	0.89	0.68–1.14	0.35	0.93	0.71–1.21	0.60
Asturias	0.96	0.73–1.26	0.79	1.14	0.86–1.50	0.36
Canarias	0.69	0.39–1.13	0.16	0.58	0.32–0.97	0.050
Cantabria	1.34	1.08–1.67	0.009	1.39	1.10–1.75	0.005
Castilla-León	1.49	1.23–1.80	<0.001	1.94	1.58–2.38	<0.001
Catalonia	1.26	1.08–1.48	0.003	1.51	1.28–1.79	<0.001
Basque country	0.69	0.46–1.01	0.07	0.81	0.53–1.19	0.30
Valencia	0.58	0.44–0.74	<0.001	0.66	0.50–0.86	0.002
TB incidence in country of origin						
Low/intermediate	Reference			Reference		
High	2.15	1.67–2.77	<0.001	3.51	2.67–4.60	<0.001
Year of TB screening	0.97	0.96–0.99	0.002	0.93	0.91–0.94	<0.001

CI: confidence interval; TB tuberculosis; IBD: inflammatory bowel disease; IMM: immunomodulator; anti-TNF: antitumor necrosis factor.

**Table 4 jcm-11-03915-t004:** Performance of screening strategies for the diagnosis of LTBI in patients with IBD.

	All Patients			Patients without IMM			Patients with IMM		
	N	LTBI, *n* (%)	OR (95% CI); *p*-Value	N	LTBI, *n* (%)	OR (95% CI); *p*-Value	N	LTBI, *n* (%)	OR (95% CI); *p*-Value
**Dual vs. single test**									
TST and IGRA	1471	342 (23)	Reference	1032	260 (25)	Reference	439	82 (19)	Reference
only TST	4557	852 (19)	0.76 (0.66–0.88); <0.001	3202	632 (20)	0.73 (0.62–0.86); <0.001	1355	220 (16)	0.76 (0.64–1.12); 0.2 *
only IGRA	1511	234 (16)	0.60 (0.50–0.73); <0.001	1176	184 (16)	0.55 (0.45–0.68); <0.001	335	50 (15)	0.84 (0.52–1.12); 0.2 *
**2 step TST vs. 1 step TST**									
1 step TST	6028	802 (13)	Reference	4234	618 (15)	Reference	1794	184 (10)	Reference
2 step TST	6028	965 (16)	1.24 (1.12–1.37); <0.001	4234	736 (17)	1.23 (1.09–1.39); <0.001	1794	229 (13)	1.28 (1.04–1.60); 0.038
**Early vs. routine screening**									
Early	4365	913 (21)	Reference	3523	758 (22)	Reference	842	155 (18)	Reference
Routine	1421	203 (14)	0.63 (0.53–0.74); <0.001	729	99 (14)	0.57 (0.45–0.72); <0.001	588	104 (15)	0.78 (0.60–1.03); 0.08 *
**Dual vs. single test in early and in routine screening**									
Early screening	4365			3523			842		
TST and IGRA	801	226 (28)	Reference	643	188 (29)	Reference	158	38 (24)	Reference
only TST	2632	535 (20)	0.65 (0.54–0.78); <0.001	2089	442 (21)	0.65 (0.53–0.79); <0.001	543	93 (17)	0.65 (0.43–1.01); 0.05 *
only IGRA	932	152 (16)	0.50 (0.39–0.62); <0.001	791	128 (16)	0.47 (0.36–0.60); <0.001	141	24 (17)	0.65 (0.36–1.14); 0.14 *
Routine screening	1421			729			692		
TST and IGRA	335	48 (14)	Reference	187	25 (13)	Reference	148	23 (16)	Reference
only TST	881	130 (15)	1.04 (0.73–1.49); 0.9	417	47 (15)	1.11 (0.68–1.86); 0.7 *	464	69 (15)	0.95 (0.58–1.61); 0.8 *
only IGRA	205	25 (12)	0.83 (0.49–1.38); 0.5 *	125	13 (10)	0.75 (0.36–1.51); 0.4 *	80	12 (15)	0.96 (0.44–2.02); 0.9 *

LTBI: latent tuberculosis infection; IMM: immunomodulator; TST: tuberculin skin test; IGRA: interferon gamma release assay. * Comparisons showing a non-significant trend in which the non-inferiority criterion was evaluated.

**Table 5 jcm-11-03915-t005:** Probability (%) of LTBI according to gender, smoking status, immunomodulator use at screening, strategy of screening (dual versus single, early versus routine) and origin in a country with a high or intermediate/low incidence of tuberculosis.

			Men						Women					
			IGRA and TST		IGRA		TST		IGRA and TST		IGRA		TST	
			No IMM	With IMM	No IMM	With IMM	No IMM	With IMM	No IMM	With IMM	No IMM	With IMM	No IMM	With IMM
Early screening	High TB incidence	Ever smoked	53	50	41	38	46	43	48	46	37	34	41	39
		Never smoked	41	38	30	28	34	32	37	34	27	25	30	28
	Low/intermediate TB incidence	Ever smoked	31	29	22	20	25	23	27	25	19	17	22	20
		Never smoked	22	20	15	13	17	16	19	17	13	12	15	14
Routine screening	High TB incidence	Ever smoked	43	41	32	30	37	34	39	37	29	26	33	30
		Never smoked	32	30	23	21	26	24	28	26	20	18	23	21
	Low/intermediate TB incidence	Ever smoked	23	22	16	15	19	17	20	19	14	13	16	15
		Never smoked	16	15	11	10	13	11	14	13	9	8	11	10

IGRA: interferon gamma release assay; TST: tuberculin skin test; IMM: immunomodulator; TB: tuberculosis.

## Data Availability

Original data from this study will be provided upon reasonable request.

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
