# Peer review of "Performance of Screening Strategies for Latent Tuberculosis Infection in Patients with Inflammatory Bowel Disease: Results from the ENEIDA Registry of GETECCU"

_jcm, 2022, doi:10.3390/jcm11133915_

Round 1

Reviewer 1 Report

The authors conducted a retrospective study of the effectiveness of latent tuberculosis infection in IBD patients.

The results described are highly relevant for clinicians treating IBD patients with immunosuppressive agents.

Author Response

Thank you for your comment: “The results described are highly relevant for clinicians treating IBD patients with immunosuppressive agents.”

With regard to your observation that, “English language and style are fine/minor spell check required”, the text has been revised by an experienced, native, professional corrector (David Bridgewater), who is thanked in the Acknowledgements section.

Reviewer 2 Report

It's a interesting and crucial study in Latent tuberculosis screening in patients with inflammatory bowel disease, especially before anti-TNF. In Table 3, we found anti-TNF at TB screening is also associated with diagnosis of latent tuberculosis infection in patients with inflammatory bowel disease. IMM use was not an independent factor.  If the authors want to conclude IMM use affected the results of LBTI screening. The patients under anti-TNF treatment should be excluded in the beginning. Furthermore, the result of univariant analysis should be provided in Table 3. Table 5 is informative and interesting. However, what is the distribution of anti-TNF user in Table 5? Most anti-TNF users also took IMM, especially infliximab. How can we know the different results in with/without IMM groups not due to anti-TNF combined use? The author should clarify this issue.   

Author Response

We would like to thank the reviewer for their comments regarding the potential bias of including patients on anti-TNF treatment at the time of latent TB screening. We believe that these patients had probably already undergone screening prior to starting anti-TNF therapy, which had not been included in the ENEIDA registry by local researchers. We have analysed again the risk factors for LTBI excluding these 369 patients, without finding changes in the results obtained in the previous analysis. We include these results as a supplementary table (Supplementary Table S1).

- In Table 3, we include the results of the univariate analysis.

- The exclusion of patients who were receiving anti-TNF, as occurred with the previously performed logistic regression analysis, did not significantly modify the results shown in Table 5 and a separate table has been created incorporating these data (Supplementary Table 3S).

Reviewer 3 Report

Review for JOURNAL OF CLINICAL MEDICINE

Manuscript Performance of screening strategies for latent tuberculosis infection in patients with inflammatory bowel disease: results 3 from the ENEIDA registry of GETECCU.

Dear doctor, thank you for the opportunity to review this interesting manuscript.
However, I have some suggestions before it can be published.

ABSTRACT

            This section is adequate; however, I suggest starting with the sentence “Patients receiving antitumor necrosis factor (anti-TNF) therapy are at risk of devel- 108 oping tuberculosis (TB), usually due to the reactivation of a latent TB infection (LTBI)” and I also suggest removing the key-word “screening strategies”.

INTRODUCTION

In lines 119-120, we see “…and the interferon-gamma release assay [IGRA])…”

However, in the abstract, we see “interferon-ץ-release assay [IGRA]”. Please, make a standardization.

Patients with Inflammatory Bowel Disease are at the heart of this study. However, there is not a sentence in the Introduction talking about the disease and what are its main forms (Ulcerative Colitis and Crohn's Disease). Please, include a paragraph on this topic.

I also suggest including newer references in this section.

METHODS

In this section, we see: “This is a retrospective, descriptive, multicenter study, conducted in patients included in the ENEIDA registry. ENEIDA is a nationwide Spanish database of IBD patients pro- moted by GETECCU [18]. The registry was approved by the Ethics Committee of each participating centre and all patients gave their informed consent. The study was approved by the ENEIDA Scientific Committee and, at the time of the data extraction (January 2018), 150 the registry included 31,827 patients.”. Please, include the names of the center involved in the study (name of hospitals, universities…). This is a piece of important information.

Please, explain what GETECCU is.

In the Abstract (lines 89-91) we see: “Methods: Patients in the Spanish ENEIDA registry with IBD screened for LTBI between 2003 and 2017 were included.” In lines 154-155 we see “…publication of the first GETECCU recommendations for LTBI screening in IBD patients) [19] to January 2018”. Did the study finish in 2017 or in January 2018?

In the title of table 2 (Table 2. Frequency of latent tuberculosis infection and results of the screening methods.). The authors already used LTBI previously in the text. Please, also use it in the title too of this table and in the other. Do the same with IBD.

RESULTS

 In table 3 we can find Geographical areas such as Madrid, Aragón, Asturias, Canarias, Cantabria, Castilla-León, Catalonia, Basque, Valencia, however, these regions were not mentioned in the Methods section. Please revise.

The font size of the letters used in table 5 is larger than those used in the text. I suggest reducing it because it will be better to visualize. I also suggest removing the bold.

DISCUSSION

I suggest adding a separate Discussion in light of the results obtained with patients with Ulcerative Colitis and with Crohn's disease. When mention is made of "The study evaluated in a real-life setting the sensitivity of early screening performed on patients for anti-TNF therapy was not foreseen compared with routine screening performed before the initiation of biological therapy, and we found that early screening markedly increased the likelihood of detecting LTBI". Is there a difference between patients with those different pathologies?

Author Response

Thank you for your comments, which has allowed us to improve the article

ABSTRACT:

- In line with your suggestion, we now begin this sentence with “Patients receiving antitumor necrosis factor (anti-TNF) therapy are at risk of developing tuberculosis (TB), usually due to the reactivation of a latent TB infection (LTBI)…”

- We have removed the keyword “screening strategies”

INTRODUCTION

- We have used the term “interferon-ץ-release assay [IGRA]” rather than “interferón-gamma release assay [IGRA]”.

- We have included a paragraph describing inflammatory bowel diseases: “Inflammatory bowel diseases (IBD) make up a group of chronic inflammatory conditions including Crohn’s Disease (CD), Ulcerative Colitis (UC) and unclassified colitis (IBD unclassified). Treatment with antitumor necrosis factor biologics (anti-TNFs) is extremely important for controlling inflammatory activity in these patients.”

- We have added two more recent references to the text:

 5.- Kumar, P.; Vuyyuru, SK.; Kante, B.; Sahu, P.; Goyal, S.; Madhu, D.; Jain, S.; Ranjan, MK.; Mundhra, S.; Golla, R.; et al.  Stringent screening strategy significantly reduces reactivation rates of tuberculosis in patients with inflammatory bowel disease on anti-TNF therapy in tuberculosis endemic region. Aliment Pharmacol Ther 2022, 55, 1431-1440.

9.- Fehily, SR.; Al-Ani, AH.; Abdelmalak, J.; Rentch, C.; Zhang, E.; Denholm, JT.; Johnson, D.; Ng, SC.; Sharma, V.; Rubin, DT.; et al. Review article: latent tuberculosis in patients with inflammatory bowel diseases receiving immunosuppression-risks, screening, diagnosis and management. Aliment Pharmacol Ther 2022, 56, 6-27.

METHODS

- The names of the centres involved in the study has been included in the supplementary Appendix.

- GETTECU is now explained in the text: “ENEIDA is a nationwide Spanish database of IBD patients promoted by GETECCU, which is a non-profit association whose mission is to improve the lives of people with IBD by promoting excellence in health care, teaching and research and influencing political and social initiatives.”

- The study was completed in January 2018 and this date has been corrected throughout the paper.

- Inflammatory bowel disease has been replaced by IBD and latent tuberculosis infection has been replaced by LTBI in the tables.

RESULTS

- We have included the names of the diverse geographical regions of Spain that participated in the study in the Methods section.

- Capital letters and bold type use have been modified in Table 5.

DISCUSSION

- Following the reviewer's recommendations, we have included a statement in the discussion section stating that we did not find differences in the performance of the different latent TB screening strategies between patients with CU and with CD: “The study evaluated in a real-life setting the sensitivity of early screening performed on patients for whom anti-TNF therapy was not foreseen compared with routine screening performed before the initiation of biological therapy, and we found that early screening markedly increased the likelihood of detecting LTBI in both UC and CD patients.”